# Optimizing Optical Dielectrophoretic (ODEP) Performance: Position- and Size-Dependent Droplet Manipulation in an Open-Chamber Oil Medium

**DOI:** 10.3390/mi15010119

**Published:** 2024-01-11

**Authors:** Md Aminul Islam, Sung-Yong Park

**Affiliations:** Department of Mechanical Engineering, San Diego State University, San Diego, CA 92182-1323, USA

**Keywords:** dielectrophoresis (DEP), optical manipulation, droplet dynamics, photoconductive

## Abstract

An optimization study is presented to enhance optical dielectrophoretic (ODEP) performance for effective manipulation of an oil-immersed droplet in the floating electrode optoelectronic tweezers (FEOET) device. This study focuses on understanding how the droplet’s position and size, relative to light illumination, affect the maximum ODEP force. Numerical simulations identified the characteristic length (*L*_c_) of the electric field as a pivotal factor, representing the location of peak field strength. Utilizing 3D finite element simulations, the ODEP force is calculated through the Maxwell stress tensor by integrating the electric field strength over the droplet’s surface and then analyzed as a function of the droplet’s position and size normalized to *L*_c_. Our findings reveal that the optimal position is *x*_opt_
*= L*_c_
*+ r*, (with *r* being the droplet radius), while the optimal droplet size is *r*_opt_ = 5*L*_c_, maximizing light-induced field perturbation around the droplet. Experimental validations involving the tracking of droplet dynamics corroborated these findings. Especially, a droplet sized at *r* = 5*L*_c_ demonstrated the greatest optical actuation by performing the longest travel distance of 13.5 mm with its highest moving speed of 6.15 mm/s, when it was initially positioned at *x*_0_
*= L*_c_
*+ r =* 6*L*_c_ from the light’s center. These results align well with our simulations, confirming the criticality of both the position (*x*_opt_) and size (*r*_opt_) for maximizing ODEP force. This study not only provides a deeper understanding of the position- and size-dependent parameters for effective droplet manipulation in FEOET systems, but also advances the development of low-cost, disposable, lab-on-a-chip (LOC) devices for multiplexed biological and biochemical analyses.

## 1. Introduction

Droplet-based microfluidic systems have drawn great interest for various lab-on-a-chip (LOC) applications [1,2,3,4]. Various chemical or biological compounds are encapsulated in discrete droplets to facilitate numerous independent reactions at an accelerated rate while minimizing the total reagent volume required [5,6,7]. Compared to traditional single-phase flow techniques [8,9], droplet-based systems offer several advantages, including rapid mixing, efficient reagent transport without dispersion, and a reduction in cross-contamination between droplets, owing to their discrete flow in an immiscible phase [10,11,12,13]. These benefits have catalyzed the development of diverse applications on droplet-based microfluidic devices, such as enzyme kinetic assays [14,15,16], polymerase chain reaction (PCR) for DNA amplification [17,18,19], and the synthesis of organic molecules or nanoparticles [20,21,22]. However, droplet-based microfluidics have difficulty, particularly in the manipulation of individual droplets driven by continuous oil flows confined in microfluidic channels. To address this, various active droplet manipulation mechanisms have been developed, including thermocapillary force [23,24,25], electrowetting on dielectric (EWOD) [26,27,28], dielectrophoresis (DEP) [29,30,31], acoustofluidic force [32,33,34], and contact charge electrophoresis (CCEP) [35,36]. Among these, DEP is notable for its low-cost, label-free approach, enabling particle manipulation through electric field non-uniformity without surface contact, thereby minimizing potential droplet–electrode contamination [37,38,39]. Various control parameters, such as the particle’s shape and size, the electrical properties of the particle and suspending medium, and the electric field frequency, allow for DEP manipulation with high flexibility and selectivity. However, DEP-based microfluidic devices often face challenges, including costly and time-consuming microfabrication processes, as well as complexities in wiring and electrode interconnection for large-scale, parallel manipulation [38,40].

In recent years, optical methods have gained prominence as cost-effective solutions to overcome challenges in wiring and interconnection in various microfluidic systems [41,42,43,44,45]. A notable example is optoelectronic tweezers (OET), which facilitate massively parallel manipulation of microscopic particles [43]. OET employ direct optical images to pattern DEP forces on a featureless photoconductive surface, enabling the manipulation of up to 15,000 particles within a 1.3 × 1.0 mm^2^ area, circumventing complex wiring issues. However, the OET application in droplet-based microfluidics encounters limitations due to impedance mismatch between the oil and photoconductive layers, restricting electric field modulation in an insulating oil medium commonly used in these systems [46,47]. To overcome this, Park et al. introduced the floating electrode optoelectronic tweezers (FEOET) mechanism [48]. By locally modifying a laterally applied electric field, the FEOET achieved optical DEP manipulation of a water droplet in an open-chamber oil medium with a remarkably low light intensity at 4.08 μW/mm^2^. This is significantly lower than traditional OET methods (four orders lower light intensity requirement), facilitating parallel manipulation of numerous droplets on cost-effective LOC chips. This FEOET technology was further advanced by using paired-diamond-shaped optical projections, demonstrating two-dimensional (2D) parallel manipulation of sixteen droplets on the FEOET’s photoconductive surface [49]. The unique configuration of the FEOET in a single-sided, open-chamber device allows seamless integration with other microfluidic components, such as micropipettes, arrayed microwells, and microchannel-based droplet generators. This integration enhances the versatility and functionality of the device for multiplexed biochemical analyses. Utilizing the FEOET principle, various microfluidic functions including droplet transport, mixing, merging, and parallel processing have been successfully demonstrated, emphasizing the simplicity and functionality of the device [49]. A later FEOET study introduced a plasmonic light scattering technique to greatly enhance optical DEP performance [50]. By incorporating metal nanoparticles, plasmonic field enhancement significantly increased the light absorption of low-quality photoconductive materials, leading to highly enhanced DEP forces for effective manipulation of oil-immersed aqueous droplets. Despite these advancements and advantages of the FEOET device, including simplicity, flexibility, and cost-effectiveness, there still remains a need for a deeper understanding of the optimal conditions that can maximize DEP forces. This understanding is crucial to fully leverage the potential of the FEOET in developing low-cost, disposable LOC chips for biological and biochemical analyses [51,52,53].

This article presents an optimization study for enhancing the optical dielectrophoretic (ODEP) performance to effectively manipulate an oil-immersed droplet on a single-sided photoconductive surface of the FEOET device. This study primarily focuses on the positional and size-dependent effects to maximize ODEP forces. The investigation begins with examining the conductivity switching performance induced by light in the FEOET device, determining the characteristic length (*L*_c_) of the electric field, defined as the location of peak field strength. Utilizing 3D finite element simulations, the ODEP force exerted on a spherical droplet was computed through the Maxwell stress tensor by integrating the electric field strength over the droplet’s surface. Our findings indicate that the optimal position for maximizing ODEP force is at *x*_opt_
*= L*_c_
*+ r*, where *r* is the droplet radius. Additionally, the droplet size was optimized to *r*_opt_ = 5*L*_c_, enhancing the light-induced field perturbation and consequently creating the largest ODEP force. Experimental validation reinforced these simulation results. A water droplet sized at *r* = 5*L*_c_ achieved the greatest movement efficiency with the longest travel distance of 13.5 mm and the highest moving speed of 6.15 mm/s when it was initially positioned at *x*_0_
*= L*_c_
*+ r =* 6*L*_c_ from the light illumination center. These experimental results confirm that optimal droplet transportation is achieved at the position of *x*_opt_
*= L*_c_
*+ r*, regardless of its size. Furthermore, a separate size-dependent investigation revealed that the maximal droplet dynamics was obtained for its optimum size of *r*_opt_ = 5*L*_c_. The study on the position- and size-dependent optimal conditions for maximizing ODEP force provides its potential applications in developing cost-effective and disposable FEOET devices for multiplexed biological and biochemical analyses.

## 2. An Overview of a Light-Driven DEP Droplet Manipulation Platform

### 2.1. Device Structure and Its Fabrication

Figure 1a presents a schematic of the FEOET device that enables ODEP manipulation of an aqueous droplet on a single-sided photoconductive surface. The device fabrication begins with patterning two indium tin oxide (ITO) electrodes on a glass substrate using a wet-etching process, creating a 30 mm gap between them at the edges of the device. A key component in this device is the photo-responsive layer, for which we utilized a polymer-based photoconductive material, titanium oxide phthalocyanine (TiOPc) [54,55]. With this polymer-based material, its thin-film layer can be easily fabricated via an inexpensive, spin-coating fabrication step without the need for high-cost and complex facilities like chemical vapor deposition (CVD) or plasma-enhanced chemical vapor deposition (PECVD) [56,57]. A TiOPc solution was prepared by dissolving its powder (Sigma-Aldrich, St. Louis, MO, USA) in a chlorobenzene solvent (1.0 wt%) at 80 °C for 2 h. This solution was then drop-casted onto the substrate and left to dry at room temperature for an additional 2 h, resulting in a 20 µm thick TiOPc layer above the ITO electrodes. However, TiOPc’s poor light absorption capability, especially when compared with semiconductor materials like amorphous silicon (a-Si) commonly used in light-driven microfluidics [41,48,49,58,59], posed a challenge. This limitation is attributed to its restricted charge carrier mobility and low exciton diffusion length at the donor/acceptor interface [60,61,62,63]. As a result, most incoming rays pass through the TiOPc layer without significantly contributing to its photoconductivity, as depicted in Figure 1b, leading to poor ODEP performance. To address this, a plasmonic-enhanced method using metallic nanoparticles was introduced [50,64]. These nanoparticles, dispersed atop the TiOPc layer, induce plasmonic scattering over a wide angular spread, effectively increasing the optical path lengths of the light rays as shown in Figure 1c. This scattering redirects more light onto the TiOPc layer, thereby substantially improving its light absorption and photoconductive performance. This enhancement enables more effective ODEP manipulation of droplets, with a demonstrated increase in actuation speed by more than 11-fold.

In this study, we retained the previously established device structure, leveraging the improved photoconductive efficiency of the TiOPc layer augmented with plasmonic light scattering effects. To achieve this, we utilized aluminum (Al) nanoparticles approximately 50 nm in diameter. Al nanoparticle powder (0220XH, Skyspring Nanomaterials, Houston, TX, USA) in 2.0 wt% was first dissolved in a Teflon solution (AF1601, DuPont, Wilmington, DE, USA). To ensure the effective dispersion of these nanoparticles, the solution was further diluted to 3% using Fluoroinert Liquid (FC-72, 3M, Saint Paul, MN, USA). This diluted solution was then spin-coated onto the TiOPc layer and cured at 110 °C for another 2 h, obtaining a 1.8 µm thick layer of the nanoparticles. Finally, an open PDMS chamber with a 25 µm thickness was placed above the nanoparticle layer to house an aqueous droplet suspended in an oil medium.

### 2.2. Droplet Actuation Principle on an FEOET Device

Droplet actuation on an FEOET device is based on the electrostatic interaction of an aqueous droplet with a light-induced, non-uniform electric field pattern created on a photoconductive surface. In the absence of light illumination, a uniform electric field is established along a lateral direction when a direct current (DC) bias voltage is applied between two ITO electrodes patterned at the edges of the device. Upon illumination, the photoconductive layer generates electron–hole pairs to alter its photo-state conductivity, essentially creating a virtual electrode at the illuminated region. Consequently, the previously uniform electric field is significantly disrupted, resulting in a non-uniform electric field pattern necessary for DEP manipulation of the droplet.

An aqueous droplet is loaded atop the TiOPc layer covered with a PDMS insulation layer. In the absence of light, an electric dipole forms across the droplet’s surface due to charge accumulation along the lateral direction. This dipole distribution is symmetrical to result in the zero net DEP force. This scenario changes significantly upon light exposure. When a light beam targets an area near one edge of the droplet, it disrupts the previously balanced electric field pattern. This illumination significantly reduces the field strength at the illuminated region, leading to an asymmetrical field distribution around the droplet. This imbalance creates a net positive DEP force, compelling the droplet to move toward the region with the stronger electric field. This light-induced alteration in the field strength and distribution is the key mechanism enabling the controlled manipulation of the droplet in the FEOET device.

## 3. Numerical Simulation Study

In this section, three-dimensional (3D) numerical simulation results are discussed to understand the optimal position and size of the droplet that can maximize ODEP force. For this simulation study, finite element software (COMSOL Multiphysics 5.6) was employed to estimate the ODEP force electrostatically exerted on a spherical droplet placed on the photoconductive surface.

### 3.1. Photoconductivity Measurement of the Nanoparticle-Coated TiOPc Layer

As described in the previous section, the nanoparticles dispersed on the TiOPc layer enhance its photoconductive performance via the plasmonic light scattering effect [50]. We faced a critical question of how to accurately model this plasmonic-enhanced photoconductive layer for numerical simulations. To address this, we undertook experimental measurements of the dark- and photo-state conductivities (*σ*_dark_ and *σ*_photo_) of the nanoparticle-coated TiOPc layer. These measurements were pivotal in characterizing the conductivity switching performance of the TiOPc layer, which will be further used to accurately define the conductivity profile of the TiOPc layer in our subsequent simulation studies.

For the conductivity measurement study, we fabricated an identical FEOET device with the modification of excluding the open PDMS chamber. The nanoparticle-coated TiOPc layer was patterned to bridge across two electrodes at the edges of the device with *L* = 1 mm in length and *W* = 4 mm in width. To assess the photoconductivity of this layer, we employed a 632.8 nm laser (HNL050LB, Thorlabs, Newton, NJ, USA) as the light source. The laser beam was further expanded by a 5× beam expander to ensure complete coverage of the 1 × 4 mm^2^ active area of the TiOPc layer. The electric resistance (*R*_TiOPc_) of the nanoparticle-coated TiOPc layer, bridged between two electrodes, was measured using a high-precision electrometer (B2985A, Keysight, Santa Rosa, CA, USA), serially connected with a 1 GΩ reference load. From the measured resistance values, we calculated the electrical conductivity (*σ*) of the nanoparticle-coated TiOPc layer using the following equation:(1)$$\sigma =\frac{L}{R_{\mathrm{TiOPc}}A}$$
where *L* and *A* are the lateral length and cross-sectional area of the TiOPc layer. Our measurements yielded specific conductivity values under different illumination conditions. Under a dark state at an illumination intensity of *I*_dark_ = 3.70 µW/cm^2^, the dark-state conductivity was recorded as *σ*_dark_ = 2.63 × 10^−9^ S/m. Conversely, during the photo state with an intensity of *I*_photo_ = 929.96 mW/cm^2^, the photo-state conductivity was measured as *σ*_photo_ = 2.65 × 10^−6^ S/m. These measurements allow us to determine the conductivity switching performance of the nanoparticle-coated TiOPc layer, quantified as the conductivity ratio between the photo and dark states, i.e., *b* = *σ*_photo_/*σ*_dark_ = 1007. In subsequent simulation studies, these empirical data will be utilized to accurately model the Gaussian conductivity distribution of the photoconductive layer.

### 3.2. Light-Induced Electric Field Distributions

In our numerical simulations, the FEOET device was modeled to be composed of three distinct layers: a 20 µm photoconductive layer, a 25 µm PDMS layer, and a 20 mm oil medium. These dimensions are in accordance with the experimental conditions. With the presence of an electric field at 100 V/mm applied between the end planes of the device, a uniform electric field was initially created along a lateral direction. To accurately model the photoconductive layer, we made several assumptions. First, we assumed that the photoconductivity does not vary along the vertical direction. Second, the radial distribution of photoconductivity was considered to be linearly proportional to the intensity profile of a Gaussian laser beam. Based on these assumptions, the photoconductive layer was assigned a two-dimensional (2D) Gaussian conductivity distribution in the simulations. This distribution is mathematically represented as follows:(2)$$\sigma =\sigma_{\mathrm{dark}}+\sigma_{\mathrm{photo}}\exp\left[-\frac{2\left(x^{2}+y^{2}\right)}{w^{2}}\right]$$
where *w* denotes a Gaussian beam radius of the laser illumination, and *σ*_dark_ and *σ*_photo_ represent the dark- and photo-state conductivities of the photoconductive layer, which were measured from the previous experiment

Figure 2a displays a cross-sectional view of the electric field distribution when subjected to illumination by a Gaussian laser beam. The beam radius was set at *w* = 0.405 mm, and the photoconductive layer was modeled based on its observed conductivity switching performance, where *b* = *σ*_photo_/*σ*_dark_ = 1007. Figure 2b further illustrates the extracted electric field strength data at various locations along the *z*-direction. Notably, the distribution is symmetric at the center of the light illumination at *x* = 0. At the surface directly above the PDMS layer (i.e., *z* = 0), a significant decrease in field strength is observed in the middle of the illuminated area, while a pronounced enhancement is seen at the illumination’s edges. As per the Gaussian conductivity distribution detailed in Equation (2), the electric field strength reaches its peak of *E* = 1.04 × 10^5^ V/m at *x* = ±0.78 mm from the center of illumination, as shown in Figure 2b. This point of the peak field strength is identified as a characteristic length of *L*_c_ = 0.78 mm. This characteristic length of the electric field plays a pivotal role in determining the optimal position and size of the droplet for maximizing the ODEP force. The implications of this characteristic length on the droplet’s optimal conditions are extensively explored in the following sections.

Figure 3 illustrates the results of three-dimensional (3D) numerical simulations depicting the field distribution when an aqueous droplet is positioned in the 10 × 4 × 2 mm^3^ oil medium of the FEOET device. The droplet is modeled as a sphere with a radius equivalent to the characteristic length, *r* = *L*_c_ = 0.78 mm. In the absence of light illumination, as depicted in Figure 3a, an electric dipole is induced, creating a strong field pattern at both edges of the droplet along the lateral direction. However, this field pattern is symmetrically balanced around the droplet, resulting in a net zero DEP force and, consequently, no movement of the droplet. To investigate the light-induced field perturbation on the droplet’s surface, we positioned the droplet at *x*_0_ = *L*_c_ from the center of the light illumination. Upon illumination, a considerable change occurs in the field distribution: the previously strong field pattern on the left side of the droplet significantly diminishes, as shown in Figure 3b. This alteration leads to an asymmetric and unbalanced field pattern around the droplet, thereby generating a net positive DEP force. This force acts to drive the droplet away from the illuminated area. The bottom two plots in Figure 3 demonstrate the distinct differences in the field distributions on the droplet’s surface under conditions without and with the light illumination at the droplet’s left side. These results clearly indicate the influence of light-induced field perturbation on the droplet’s surface for ODEP manipulation on the FEOET device.

### 3.3. ODEP Force Calculation

We further focused on calculating the ODEP force exerted on the droplet. Generally, the DEP force acting on a particle can be estimated using the equation *F* = (*p* · ∇)*E*, where *p* is the field-induced dipole moment and *E* is the electric field [65,66]. This approximation holds true predominantly when the particle size is substantially smaller than the characteristic length of the electric field gradient [67,68]. However, in our case, this conventional approximation cannot be satisfied for the DEP force estimation, as the size of the droplet is comparable to the scale of the light-induced electric field gradient. To circumvent this limitation and ensure precise calculation of the DEP force in our context, we employed the Maxwell stress tensor method. Due to the DC bias operation in the FEOET, the electric field strength inside the droplet effectively becomes zero. This occurs as a result of field screening, where charges accumulate on the droplet’s surface, thereby nullifying any tangential component of the electric field on the droplet’s surface. The Maxwell stress tensor can be simplified to the following form [69]:(3)F→DEP=∫surface12εE2ds→
where *ε* is the permittivity of the surrounding medium and *E* is the electric field strength on the droplet surface.

Using the 3D finite element simulation data, we conducted a comprehensive calculation of the electric field strength acting upon each infinitesimal area of the droplet’s surface. This intricate process involved integrating these individual field strength values over the entire surface area of the droplet, utilizing spherical coordinates for precise force calculation. This approach not only provides a qualitative understanding of the ODEP force electrostatically exerted on a spherical droplet, but also helps in determining the optimal position and size of the droplet to create the maximal ODEP force.

### 3.4. A Position-Dependent Effect

We first focused on examining how the position of the droplet affects the ODEP force. To explore this position-dependent effect, we repeated the simulation in Figure 3, keeping the droplet size constant at *r* = *L*_c_ = 0.78 mm and varying only its position (*x*_0_) relative to the light illumination, as depicted in the inset of Figure 4. All other simulation parameters were maintained as previously established. Using Equation (3), we calculated the *x*-directional ODEP force (*F*_x_). The data plotted in Figure 4 show the variation in *F*_x_ with the droplet’s position (*x*_0_) from the center of the light illumination. When the droplet is centrally located within the illuminated area, i.e., *x*_0_ = 0, the field distribution around the droplet is symmetric and balanced, leading to a negligible DEP force (i.e., *F*_x_ ≈ 0). As the droplet is positioned away from the light illumination, we observed an increasing trend in *F*_x_. The force reaches its peak of *F*_x_ = 2.21 nN when the droplet is positioned at *x*_0_ = (*L*_c_ + *r*) = 1.56 mm. At this location, the one edge of the droplet aligns with the characteristic length of *L*_c_, where *F*_x_ is maximized. Beyond this point, as the droplet is located further away from the light illumination, *F*_x_ gradually decreases and eventually diminishes. It is worthwhile to note that, when the droplet is positioned to the left of the light illumination, it experiences the DEP force in the opposite direction, but with the same magnitude.

### 3.5. A Size-Dependent Effect

In our continued efforts to understand the influence of droplet size on the ODEP force, the simulation study was extended to include droplets with varying radii, ranging from *L*_c_ ≤ *r* ≤ 9*L*_c_. The results of these simulations are depicted in Figure 5. The first notable observation is that the position-dependent *F*_x_ exhibits a consistent trend across all different droplet sizes. Regardless of its size, *F*_x_ ≈ 0 at *x*_0_ = 0. This negligible force value is attributed to the symmetric and balanced field distribution around the droplet when it is centrally located within the light illumination. As its position is away from the illumination, *F*_x_ increases, reaches its peak, and then decreases to zero.

Another intriguing finding is the identification of an optimal droplet size that maximizes *F*_x_. The simulations revealed that a droplet with its size of *r* = 5*L*_c_ generates the largest DEP force, peaking at *F*_x_ = 15.89 nN when positioned at *x*_0_ = (*L*_c_ + *r*) = 4.68 mm, as shown by the red curve in Figure 5. For droplets either smaller or larger than *r* = 5*L*_c_, the maximum value of *F*_x_ decreases. This diminishing trend in *F*_x_ is attributed to the electric field distribution, which is less effectively perturbed by the light illumination in cases where the droplet size deviates from this optimal value.

### 3.6. Optimal Position and Size of the Droplet

The findings from our previous simulation studies have laid the groundwork for determining the optimal position (*x*_opt_) and size (*r*_opt_) of a droplet to maximize the ODEP force in the FEOET device. For the purpose of this optimization study, we introduce a new approach to present the droplet’s position and size in normalized forms as follows:(4)$$NP=\frac{x_{0}}{L_{c}+r}$$
(5)$$NS=\frac{r}{L_{c}}$$
where *NP* and *NS* represent the normalized position and the normalized size of the droplet, respectively. This normalization process is a critical step in our analysis, as it allows us to systematically evaluate the impact of the droplet’s position and size in relation to the characteristic length (*L*_c_). By expressing these two control parameters in a normalized form, we can more effectively compare and understand the influence of the droplet’s position and size on the ODEP force.

The comprehensive position- and size-dependent simulation results, initially presented in Figure 5, have been re-analyzed and re-plotted in the normalized terms. These re-plotted data are presented in Figure 6a and Figure 6c, respectively. At *NP* = 0, where the droplet is centrally positioned at *x*_0_ = 0 under the light illumination, the magnitude of the *x*-directional DEP force is negligible, i.e., *F*_x_ ≈ 0. This is attributed to the symmetric and balanced field distribution around the droplet. Another significant finding is the consistent attainment of the maximal *F*_x_ given at *NP* = 1 across all droplet sizes. At *NP* = 1, Equation (4) can be reformulated to define the optimal position of the droplet as *x*_opt_ = (*L*_c_ + *r*). This position corresponds to the point where the light-induced field perturbation on the droplet’s surface is maximized, thereby generating the largest *F*_x_. In other words, the results suggest that the ODEP force reaches its maximum when one edge of the droplet aligns with the characteristic length of *L*_c_, as illustrated in Figure 6b, regardless of the droplet’s size.

To further understand the size-dependent effect on the ODEP force, we conducted simulations where the droplet’s size was varied, while its position was fixed at three different locations of *NP* = 0.5, 1.0, and 1.5. The study results are presented in Figure 6c, which depicts the size-dependent *F*_x_ as a function of the normalized size (*NS*) defined in Equation (5) as the ratio of the droplet size to the characteristic length of *L*_c_. A critical observation from the graph is the behavior of *F*_x_ in relation to the *NS* value. We noted that *F*_x_ increases rapidly for values of *NS* < 5. A peak in *F*_x_ is achieved precisely at *NS* = 5 for all *NP* values. This peak in the graph indicates the optimal size of the droplet, *r*_opt_ = 5*L*_c_, which corresponds to the largest attainable *F*_x_. When *NS* exceeds 5, a decreasing trend is interestingly observed in *F*_x_. This reduction in force indicates that the light-induced field perturbation on the droplet’s surface becomes weakened as the droplet size grows beyond the optimal *NS* value. These insights into the position- and size-dependent ODEP force significantly enhance our understanding of droplet manipulation within FEOET systems.

## 4. Experimental Study

To experimentally validate the insights gleaned from our previous simulation studies regarding position- and size-dependent ODEP forces, we undertook a series of experimental studies. In these experiments, a deionized water droplet was placed into an oil-filled PDMS chamber. An electric field of 70 V/mm was uniformly applied across the entire device in a lateral direction. To induce the DEP force necessary for optical droplet manipulation, a Gaussian laser beam, with a beam radius of 0.405 mm and a total power output of 5 mW, was projected onto the nanoparticle-coated TiOPc layer. The movement of the droplet, triggered by the ODEP force, was meticulously recorded from its initial position until it came to a complete stop. This recording was achieved using a high-speed camera (VEO-E 340L, Phantom, Wayne, NJ, USA), allowing for detailed analysis of the droplet’s motion.

Figure 7 captures key moments in the ODEP actuation of an oil-immersed water droplet with its volume of 248.5 μL (equating to its radius of *r* = 5*L*_c_ = 3.9 mm) on a single-sided photoconductive surface. The droplet was initially positioned at *x*_0_ = (*L*_c_ + *r*) = 6*L*_c_ = 4.68 mm from the center of the light illumination, as shown in Figure 7b. Prior to the laser beam’s illumination, no movement of the droplet was observed, despite the application of a bias voltage (Figure 7a). This changed dramatically when the laser beam was introduced to illuminate the edge of the droplet. Upon illumination, the droplet was optically actuated, moving away from the illuminated region, as captured in Figure 7b. The droplet’s speed progressively increased during its movement. A significant observation was made at *t* = 1.8 s, where the droplet’s instantaneous speed reached its peak of 6.15 mm/s as it moved through the location at *x* = 12.49 mm (Figure 7c). Following this peak, the droplet’s speed gradually decreased until it eventually came to a stop at a final location of *x*_f_ = 18.18 mm (Figure 7d). The total travel distance of the droplet was calculated as ∆*x* = (*x*_f_ - *x*_0_) = 13.5 mm, completed over a duration of ∆*t* = 5.2s. It is worthwhile to note that we did not observe any light-induced thermal effect on droplet dynamics in the experiments.

In the following sections, we will delve into a detailed experimental study of droplet dynamics to validate the findings from our simulations on the position- and size-dependent effects on the ODEP force.

### 4.1. A Position-Dependent Experimental Study

Based on the previous simulation results in Figure 6a, we recognized that the largest ODEP force could be generated when a droplet is positioned at *NP* = 1, corresponding to the optimal position of *x*_opt_ = (*L*_c_ + *r*) away from the light illumination. This optimal position aligns one edge of the droplet with the characteristic length of *L*_c_, where the strongest field perturbation and, consequently, the maximum ODEP force can be generated. With this largest ODEP force, the droplet is expected to undergo the longest travel transportation with the highest speed.

To experimentally validate these findings, we conducted a droplet dynamics study with the normalized droplet size fixed at *NS* = 5 (i.e., *r* = 5*L*_c_). The initial position of the droplet was varied in 4*L*_c_ ≤ *x*_0_ ≤ 8*L*_c_, corresponding to 0.67 ≤ *NP* ≤ 1.33. Figure 8a presents the profile of the droplet’s instantaneous speed measured at various locations. For a comparative study on the position-dependent effect, the *x*-axis was represented as the droplet’s location relative to its initial position, i.e., (*x* − *x*_0_). Upon illumination by a Gaussian laser beam, the droplet commenced movement, reaching its peak speed before coming to a complete stop. Notably, the longest travel distance, ∆*x* = (*x*_0_ − *x*_f_) = 13.5 mm, and the highest speed, 6.15 mm/s, were observed when the droplet was initially positioned at *NP* = 1. This observation aligns well with the simulation results where the optimal position was given as *x*_opt_ = (*L*_c_ + *r*) = 6*L*_c_ for the droplet with its size of *r* = 5*L*_c_, at which the largest *F*_x_ was obtained. For the droplets initially positioned away from this optimum position in either *x* < *x*_opt_ or *x* > *x*_opt_, the ODEP force becomes weakened. Correspondingly, shorter transportation and a slower moving speed are observed, as shown in Figure 8a.

A subsequent experiment was repeated for a smaller droplet at *NS* = 1 (i.e., *r* = *L*_c_). The observed speed trend—increasing, peaking, then decreasing—was consistent (as shown in Figure 8b). Even for the smaller droplet size, the longest travel distance as ∆*x* = (*x*_f_ − *x*_0_) = 5.73 mm and the highest speed as 2.05 mm/s were achieved when the droplet was initially positioned at *NP* = 1, confirming the optimal position of *x*_opt_ = (*L*_c_ + *r*) = 2*L*_c_ = 1.56 mm at which the largest *F*_x_ was attained.

Considering the numerical and experimental study results we obtained, it becomes evident that the maximum ODEP force *F*_x_ is generated when a droplet is initially positioned at *x*_0_ = *x*_opt_ = (*L*_c_ + *r*) from the light illumination. These findings not only validate our simulation predictions, but also provide crucial insights for optimizing ODEP force in FEOET devices for practical applications.

### 4.2. A Size-Dependent Experimental Study

In continuation of our experimental efforts, we conducted a size-dependent study to further investigate the influence of droplet size on optical DEP force. In these experiments, the droplet’s initial position was consistently set at the optimal *NP* = 1, i.e., *x*_0_ = *x*_opt_
*=* (*L*_c_ + *r*), while varying the droplet size in the range of 1 ≤ *NS* ≤ 9. The instantaneous speed of the droplet was measured during its optical transportation. Figure 9a presents these speed measurements as a function of the droplet’s relative location of (*x* − *x*_0_). Aligning with our prior simulation results in Figure 6c, the experimental data confirmed that the largest *F*_x_ is generated for a droplet at *NS* = 5. Consistently, the droplet with this optimal size of *r*_opt_ = 5*L*_c_ = 3.9 mm achieved the longest travel distance as ∆*x* = (*x*_0_ − *x*_f_) = 13.5 mm and the highest speed as 6.15 m/s, as indicated by the red curve in Figure 9a. Droplets deviating from this optimal size, whether larger or smaller than NS = 5, exhibited shorter travel distances and decreased peak speeds. To further corroborate our findings on the optimal droplet size, the same experiment was replicated but with the droplets positioned at a non-optimal location of *NP* = 0.5. As illustrated in Figure 9b, even at this non-optimal position, the droplet sized at *r*_opt_ = 5*L*_c_ consistently demonstrated the longest travel distance and the highest speed, validating our simulation predictions.

Our study challenges the conventional understanding that DEP force is proportional to the cube of a particle’s radius, an approximation typically valid for particles significantly smaller than the characteristic length of the electric field [70,71]. In our case, the droplet size is comparable to the characteristic length, leading to the identification of an optimal size that maximizes the DEP force. For droplets larger than 5*L*_c_, the light-induced field perturbation becomes more localized, resulting in a diminished DEP force.

## 5. Conclusions

This study successfully conducted an in-depth optimization analysis to enhance the ODEP performance in the FEOET system. A critical aspect of this study was the investigation of the position- and size-dependent effects on ODEP forces, aimed at identifying the optimal conditions for effective droplet manipulation. Our initial approach involved analyzing the light-induced conductivity switching performance of the FEOET device to determine the characteristic length (*L*_c_) of the electric field, which corresponds to the peak field strength location. Utilizing 3D finite element simulations, we were able to calculate the ODEP force using the Maxwell stress tensor. This calculation integrated electric field strength across the surface of a spherical droplet, presenting the ODEP force as a function of both the droplet’s position and size relative to the characteristic length (*L*_c_). The findings from these simulations indicated that the optimal droplet position for maximizing ODEP force is *x*_opt_
*= L*_c_
*+ r*. Furthermore, a size-dependent study revealed that the optimal droplet size for maximizing this force is *r*_opt_ = 5*L*_c_, where the light-induced field perturbation on the droplet is at its peak. To validate the findings from these simulations, we conducted a series of droplet dynamics experiments. These experiments demonstrated that a droplet sized at *r* = 5*L*_c_ achieved the longest travel distance (13.5 mm) and the highest speed (6.15 mm/s) when positioned at *x*_opt_
*=* (*L*_c_
*+ r*) *=* 6*L*_c_. These experimental findings corroborated our simulation results, confirming that the optimal positioning and sizing of droplets are critical for maximizing light-actuated droplet transportation. This optimization study not only enhances our understanding of the position- and size-dependent factors that maximize ODEP force, but also underscores the potential of these findings in developing cost-effective, disposable, lab-on-a-chip (LOC) devices. The insights gained from this study are instrumental to advancing multiplexed biological and biochemical analyses, paving the way for more efficient and precise microfluidic applications.

## Figures and Tables

**Figure 1 micromachines-15-00119-f001:**
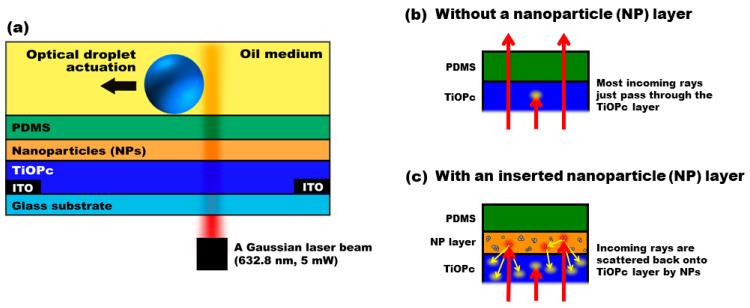
**A schematic illustration of the FEOET device**. (**a**) The device consists of a 20 µm TiOPc layer coated with a 1.8 µm nanoparticle layer, on top of which an open PDMS chamber with a 25 µm thickness is placed to house an aqueous droplet surrounded by an oil medium. (**b**) Due to the TiOPc’s poor photoconductive properties, only a small portion of the input rays can be absorbed in the TiOPc layer. As indicated by the arrows, most of the rays just pass through it before making a contribution to its photo-state conductivity, resulting in poor ODEP performance. (**c**) To enhance light absorption of the TiOPc layer, plasmonic nanoparticles were used as light scattering elements above the TiOPc layer. With the presence of metal nanoparticles, the transmitted rays (which did not previously contribute to the photo-state conductivity) underwent plasmonic light scattering. They were then re-directed onto the TiOPc layer with increased optical path lengths to effectively enlarge light absorption onto the TiOPc layer. As a result, its photo-state conductivity was greatly enhanced in the light-illuminated area, while its dark-state conductivity did not undergo much modification in the area illuminated by the dark pattern. The conductivity switching capability of the TiOPc is significantly enlarged, leading to larger ODEP forces.

**Figure 2 micromachines-15-00119-f002:**
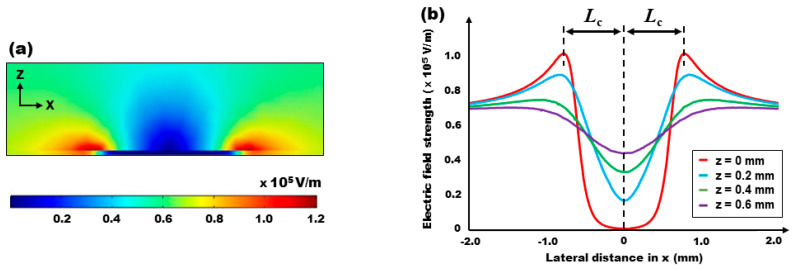
**3D numerical simulation results** showing (**a**) a cross-sectional view of the electric field distribution created by the illumination of a Gaussian laser beam and (**b**) the electric field strength data extracted at various *z*-directional locations. At the surface right above the PDMS layer (i.e., *z* = 0), the field strength greatly decreases in the middle of the illuminated area, while it is strongly enhanced around the edges of the illumination. The field strength peaks as *E* = 1.04 × 10^5^ V/m at *x* = ±0.78 mm away from the center of the light illumination. This location of the peaked field is defined as a characteristic length of *L*_c_ = 0.78 mm for the following optimization studies. As further increasing *z*-directional locations, the field gradient dramatically decreases.

**Figure 3 micromachines-15-00119-f003:**
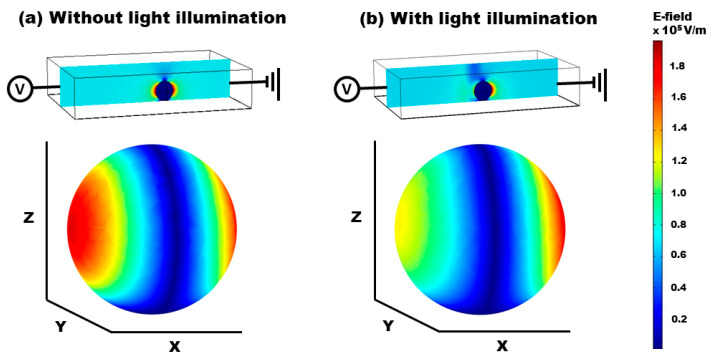
**3D numerical simulation results** showing cross-sectional views of the electric field distributions around an aqueous droplet suspended in an oil medium (**a**) without and (**b**) with the illumination of a Gaussian laser beam. The bottom two plots clearly show the significant difference in the field distributions created on the droplet’s surface without and with the light illumination at the left edge of the droplet.

**Figure 4 micromachines-15-00119-f004:**
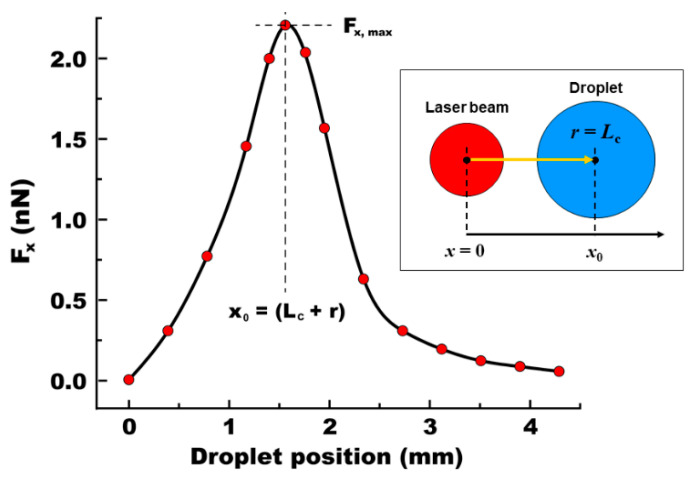
**Droplet’s position-dependent DEP force**. For simulation, the droplet radius was fixed as *r* = *L*_c_. An *x*-directional DEP force (*F*_x_) was estimated while varying its position (*x*_0_) away from the center of the light illumination, as indicated by the arrow in an inset image. At *x*_0_ = 0, the field distribution is symmetric and balanced around the droplet, resulting in *F*_x_ = 0. As its position is away from the light illumination, the force increases and reaches its peak at *F*_x_ = 2.21 nN at *x*_0_ = (*L*_c_ + *r*) = 1.56 mm. Then, the force decreases to zero as the droplet is positioned far away.

**Figure 5 micromachines-15-00119-f005:**
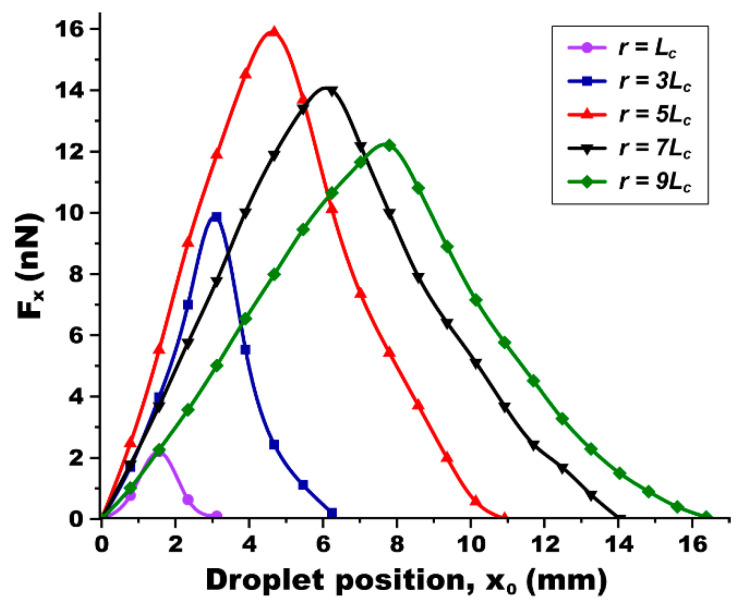
**Effects of the droplet’s position and size on the DEP force**. An *x*-directional DEP force (*F*_x_) was estimated for the droplet varied with its position and size. No matter the droplet size, a position-dependent force shows the trend increasing, being peaked, and decreasing. A size-dependent study presents the existence of the optimum droplet size as *r* = 5*L*_c_ for which *F*_x_ can be maximized as high as 15.89 nN. This maximum value in *F*_x_ decreases for both smaller and larger sizes of the droplet than *r* = 5*L*_c_.

**Figure 6 micromachines-15-00119-f006:**
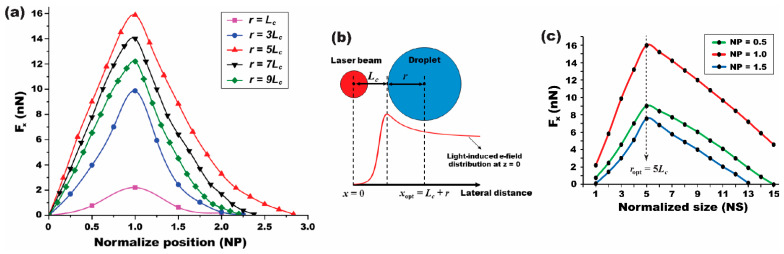
**A light-induced DEP force presented as a function of the normalized position (NP) and the normalized size (NS) of the droplet**. (**a**) An *x*-directional DEP force (*F*_x_) presented as a function of the normalized position (*NP*) of the droplet. At *NP* = 0 (i.e., *x* = 0), the field distribution is symmetric and balanced around the droplet, regardless of its size, leading to *F*_x_ = 0. An increasing trend is shown for *F*_x_ with the *NP*. The peaked force is obtained at *NP* = 1. When *NP* > 1, a decreasing trend in *F*_x_ is observed. (**b**) A schematic illustration shows the optimum position of the droplet given as *x*_opt_ = *L*_c_ + *r* when one edge of the droplet aligns with the characteristic length of *L*_c_, leading to the largest *F*_x_. (**c**) *F*_x_ was presented as a function of the normalized size (*NS*) of the droplet. *F*_x_ rapidly increases when *NS* < 5. Then, the peaked *F*_x_ was obtained at *NS* = 5, no matter what the *NP* values are. Thus, the optimum size of the droplet can be found as *r*_opt_ = 5*L*_c_ for which the light-induced field perturbation can be maximized to lead to the largest *F*_x_. For *NS* > 5, a decreasing trend is observed for *F*_x_.

**Figure 7 micromachines-15-00119-f007:**
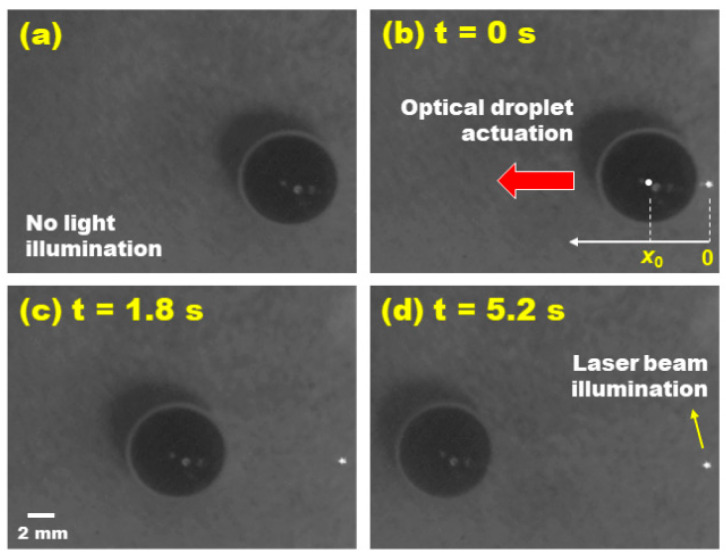
**Video snapshots of the ODEP-driven droplet transportation**. A dye-added water droplet was initially positioned at *x*_0_ = (*L*_c_ + *r*) = 4.68 mm away from the center of the light illumination. (**a**) A bias voltage was applied between two electrodes patterned on both edges of the device, but no droplet actuation was induced without the light illumination. (**b**) When a laser beam was projected at the edge of the droplet, it was optically actuated to move away from the light illumination as indicated by the red arrow. (**c**) At *t* = 1.8s, the droplet reaches its highest speed of 6.15 mm/s. (**d**) Its transportation was completed at its final position of *x*_f_ = 18.18 mm.

**Figure 8 micromachines-15-00119-f008:**
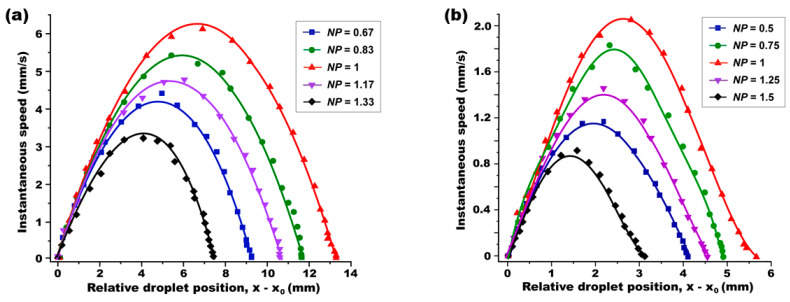
**A position-dependent experimental study for the droplet whose size was fixed as (a) *r* = 5*L*_c_ and (b) *r* = *L*_c_.** A light-actuated droplet dynamics study was carried out while its initial position (*x*_0_) was variously tuned from 4*L*_c_ to 8*L*_c_ away from the center of the light illumination. For both cases, the longest transportation distance and the highest speed were obtained for the droplet when it was initially positioned at *NP* = 1. This observation therefore indicates the optimal position is given as *x*_opt_ = (*L*_c_ + *r*), at which the largest *F*_x_ can be accomplished.

**Figure 9 micromachines-15-00119-f009:**
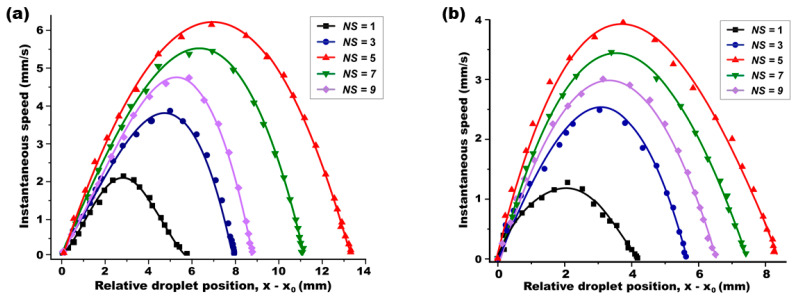
**A size-dependent experimental study in which the droplet is initially located at (a) the optimal position of *NP* = 1 and (b) the non-optimum position of *NP* = 0.5.** A light-actuated droplet dynamics study was carried out while its normalized size (*NS*) was variously tuned. For both cases, the longest transportation distance and the highest speed were obtained for the droplet when it was initially positioned at *NP* = 1. This observation therefore indicates the optimum position is given as *x*_opt_ = (*L*_c_ + *r*), at which the largest *F*_x_ can be accomplished.

## Data Availability

Data is contained within the article.

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
