# Peer review of "Optimizing Optical Dielectrophoretic (ODEP) Performance: Position- and Size-Dependent Droplet Manipulation in an Open-Chamber Oil Medium"

_micromachines, 2024, doi:10.3390/mi15010119_

Round 1

Reviewer 1 Report

Comments and Suggestions for Authors

The authors present an optimization study to enhance optical dielectrophoretic (ODEP) performance for effective manipulation of an oil-immersed droplet in the floating electrode optoelectronic tweezers (FEOET) device. This work is well done and well written. I only have concerns on the force calculation in Eq 3. 
Are the particle's own dielectric properties accounted for? It seems Eq 3 simplified with limited applicability.

Author Response

Reviewer #1: The authors present an optimization study to enhance optical dielectrophoretic (ODEP) performance for effective manipulation of an oil-immersed droplet in the floating electrode optoelectronic tweezers (FEOET) device. This work is well done and well written.

Comment #1: I only have concerns on the force calculation in Eq 3. Are the particle's own dielectric properties accounted for? It seems Eq 3 simplified with limited applicability.

Response to Comment #1: We sincerely thank the reviewer for positive comments.

In our research, we focus on an aqueous droplet submerged in a non-conductive oil medium, a configuration frequently utilized in droplet-based microfluidic systems. In such systems, the electrical conductivity of the droplet significantly surpasses that of the oil. Upon applying a DC bias voltage to the FEOET device, ion redistribution occurs within the droplet, leading to ion accumulation at the droplet's surface. This phenomenon results in field screening, effectively nullifying the electric field within the droplet. Consequently, the electric field is oriented perpendicular to the droplet's surface, with no tangential electric field components present on the surface. Based on this, Eq. (3) presents a simplified form of the Maxwell stress tensor, incorporating only the permittivity of the oil medium and the electric field impacting the droplet's surface. For our study, it is not necessary to consider the electric properties of a dielectric particle when calculating the DEP force.

Reviewer 2 Report

Comments and Suggestions for Authors

In this article, the authors thoroughly investigated optimal conditions for effective manipulation of an oil-immersed droplet in the floating electrode optoelectronic tweezers device using numerical and experimental approaches. The article is well-organized and written, and the results are clear. Furthermore, the collected data is properly analyzed and presented. It is also significant both academically and practically, and thus may be of interest to many potential Journal readers. As a result, I believe this paper can be published with minor revisions.

(1) As the authors are aware, an alternative technique known as contact charge electrophoresis (Langmuir 34 (22), 6315-6327; Langmuir 2022 38 (18), 5759-5764; Micromachines 13 (4), 593) exists alongside the approach proposed by the authors in the realm of digital microfluidics (DMF). It would be beneficial to examine and highlight the benefits of the authors' proposed method in comparison to other DMF techniques, including this alternative approach.

(2) The physical properties of the oil, such as density, dielectric constant, and viscosity, are expected to have a significant effect on the critical driving conditions or the speed of the droplets. Unfortunately, I have not been able to find any relevant information about the oil. Therefore, please provide further information on the type of oil used in this study.

(3) As supplementary material to help readers understand the study, I suggest authors include a figure showing the geometry of the numerical model and giving basic information about it (e.g. mesh, boundary condition, and governing equation).

(4) Are there any adverse heat-related effects, such as convection flow?

Author Response

A response letter was attached. 
